

# Nitrogen fertilizer amount has minimal effect on rhizosphere bacterial diversity during different growth stages of peanut

Zheng Yang[1], Lin Li[2], Wenjuan Zhu[1], Siyuan Xiao[1], Siyu Chen[1], Jing Liu[1], Qian Xu[1], Feng Guo[3] and Shile Lan[1]

[1] College of Bioscience and Biotechnology, Hunan Agricultural University, Changsha, Hunan, China
[2] College of Agronomy, Hunan Agricultural University, Changsha, Hunan, China
[3] Biotechnology Research Center, Shandong Academy of Agricultural Sciences, Jinan, Shandong, China

## ABSTRACT

The impact of short-term nitrogen fertilizer input on the structure and diversity of peanut rhizosphere microbiota (RM) at different growth stages (GSs) was explored in the southern paddy soil planting environment. Three levels of nitrogen were applied in the field: control (LN, 0 kg/hm$^2$), medium nitrogen (MN, 55.68 kg/hm$^2$), and high nitrogen (HN, 111.36 kg/hm$^2$). The rhizosphere soil was collected during four GSs for high-throughput sequencing and chemical properties analysis. The effect of nitrogen fertilizer application on peanut RM was minimal and was obvious only at the seedling stage. In the four peanut GSs, a significant increase in relative abundance was observed for only one operational taxonomic unit (OTU) of *Nitrospira* under HN conditions at the seedling stage and mature stage, while there was no consistent change in other OTUs. The difference in RM among different peanut GSs was greater than that caused by the amount of nitrogen fertilizer. This may be due to the substantial differences in soil chemical properties (especially alkali-hydrolyzable nitrogen, pH, and available potassium or total potassium) among peanut GSs, as these significantly affected the RM structure. These results are of great value to facilitate deeper understanding of the effect of nitrogen fertilizer on peanut RM structure.

## INTRODUCTION

Nitrogen, one of the major elements required by plants, is an important component of nucleotides, proteins, phospholipids, and some plant hormones. N$_2$ in its inert form cannot be absorbed and utilized by plants, so nitrogen usually becomes a restrictive nutrient element for plant growth (*Moreau et al., 2019*). The nitrogen fertilizer most widely used in agriculture is urea. Amide nitrogen is transformed into ammonium nitrogen by the action of soil urease. After absorption by plants, ammonium nitrogen is assimilated into glutamate and glutamine by the action of glutamate synthase and glutamine synthase, and then further transformed into aspartate and asparagine. Finally, various amino acids are

Corresponding authors
Feng Guo, guogeng08-08@163.com
Shile Lan, 875540378@qq.com

formed during plant growth and development (*Lea, Robinson & Stewart, 1990*; *Okumoto & Pilot, 2011*; *Hildebrandt et al., 2015*).

Rhizosphere microbiota (RM) is an important medium for material exchange between plants and soil. The number of microorganisms in rhizosphere soil is several to dozens of times higher than that in non-rhizosphere soil (*Zamioudis & Pieterse, 2012*). The composition of plant RM is affected by many biological and abiotic factors. Different soil types and their physical and chemical properties macroscopically affect RM (*Walters et al., 2018*). Soil pH, water content, carbon nitrogen ratio (C/N), nutrient content, and soil fertility are correlated with specific RMs (*George et al., 2006*; *Rousk et al., 2010*; *Philippot et al., 2013*). The species, growth stage (GS), and genotype of plants determine which microorganisms can enrich in RM (*Xu et al., 2009*; *Bouffaud et al., 2014*; *Pérez-Jaramillo, Mendes & Raaijmakers, 2015*). For example, the RM composition of low-starch and high-starch genotype potatoes is very different. The RM of low-starch genotype potato includes significantly higher relative abundances of *Sphingobium*, *Pseudomonas*, *Acinetobacter*, *Stenotrophomonas*, and *Chryseobacterium* than that of high-starch genotype potato (*Marques et al., 2014*). A study on the RM of maize showed that the relative abundances of *Massilia*, *Flavobacterium*, and *Arenimonas* were highest in the early GS, while the relative abundances of *Burkholderia*, *Ralstonia*, *Sphingobium*, *Bradyrhizobium*, and *Variovorax* were highest in the late GS (*Li et al., 2014*). Moreover, plant root exudates such as amino acids, sugars, fatty acids, vitamins, growth hormones, and organic acids are also used as chemical inducers to attract rhizobia, improve the expression of *nod* genes, and increase nodulation (*Hassan & Mathesius, 2012*).

Peanut (*Arachis hypogaea*) is an annual leguminous herb with an annual output of more than $3 \times 10^7$ kg and is rich in a variety of amino acids, vitamins, and large amounts of fat and protein. Peanuts are believed to promote cell development, improve intelligence, prevent premature senility, prevent cardiovascular and cerebrovascular diseases, and have anticoagulant and anti-aging functions (*Arya, Salve & Chauhan, 2016*; *Toomer, 2018*). Therefore, peanut has become one of the four largest oil crops worldwide (*Arya, Salve & Chauhan, 2016*). As a legume, the yield and quality of peanuts are closely related to nitrogen fertilizer application and RM. Although our previous study showed that nitrogen fertilizer application significantly improved the yield and oleic acid and linoleic acid contents of peanuts (*Yang et al., 2021*), the impact of nitrogen fertilizer application on the composition of peanut RM and the GS of this impact have not been fully described. To further elucidate the effect of nitrogen fertilizer application on the composition of peanut RM, and the GS of the effect, we analyzed the composition of peanut RM at different GSs using high-throughput sequencing.

## MATERIALS AND METHODS

### Brief description of experimental site

The experimental site was located at the Yunyuan Base of Hunan Agricultural University (27°11′ N, 113°4′ E) at an altitude of 50 m. The area has a subtropical monsoon climate, with an annual average temperature of 17.2 °C, and an average annual precipitation of 1,361.6 mm.

The experimental soil was paddy soil developed from Quaternary red soil. The pH of the soil was approximately 6.0. The levels of organic matter, total nitrogen (TN), total phosphorus (TP), total potassium (TK), alkali-hydrolyzable nitrogen (AN), available phosphorus (AP), and available potassium (AK) in the soil were 19.45, 1.35, 1.12, 29.16, 0.13, 0.07, and 0.26 g/kg, respectively.

## Experimental design

To test the effect of nitrogen fertilizer amount on peanut RM, three nitrogen levels: no nitrogen (LN), medium nitrogen (MN, 55.68 kg/hm$^2$), and high nitrogen (HN, 111.36 kg/hm$^2$), were applied with 31.42 kg/hm$^2$ of phosphate fertilizer (calculated as P) and 142.45 kg/hm$^2$ of potassium fertilizer (calculated as K) 2 days before sowing. The tested fertilizers were urea (nitrogen ≥ 464 g/kg), calcium magnesium phosphate ($P_2O_5$ ≥ 120 g/kg), and potassium sulfate ($K_2O$ ≥ 520 g/kg, Cl ≤ 15 g/kg, S ≥ 175 g/kg). After ridging, fertilizer was evenly sprinkled on the ridge and turned into the 0–20 cm depth soil layer.

Three replicates of each nitrogen application level were arranged randomly (Fig. 1A). Each experimental subarea was 5 × 4 m (Fig. 1A). There were five ridges in each experimental subarea. The ridge width and height were 80 and 12 cm, respectively. Row spacing of the ridge was 25 cm. The plant spacing was 11 cm, and the theoretical density was 227,280 plants/hm$^2$.

The tested peanut was the big peanut variety Jihua 16 with a high oleic acid content (80.1%), which was purchased from the Institute of Grain and Oil Crops, Hebei Academy of Agricultural and Forestry Sciences. The peanuts were sown on April 28 and harvested on September 2, 2019.

## Sample collection

At the seedling stage (SS, 38 days after sowing), flower peg stage (FPS, 70 days after sowing), pod setting stage (PSS, 95 days after sowing), and mature stage (MS, 120 days after sowing; Figs. 1B–1D), six peanut plants in each subarea in the normal growth area without missing seedlings were randomly selected and dug up and large pieces of soil on the root surface were shaken off. Then the rhizosphere soil of peanut plants was collected into plastic bags, mixed well, and taken back to the laboratory for determination of soil physical and chemical properties.

The rhizosphere soil of peanut plants was also collected into sterile plastic bags, mixed well, quick-frozen in liquid nitrogen, and then stored at −80 °C until microbiota DNA extraction.

## Chemical analysis

The sample solution was prepared according to a water–soil ratio of 1:2.5, and the soil pH was measured using a pH meter. The soil TN content was measured using a DigiPREP TKN system (SCP, Vaughan, Canada). The soil TP content was determined using NaOH melting-molybdenum antimony colorimetry (Jin et al., 2020). The soil melt was prepared by melting with NaOH and then diluted, and the soil TK content was determined using a flame photometer (Stanford & English, 1949; Blanchet et al., 2017). The soil AN

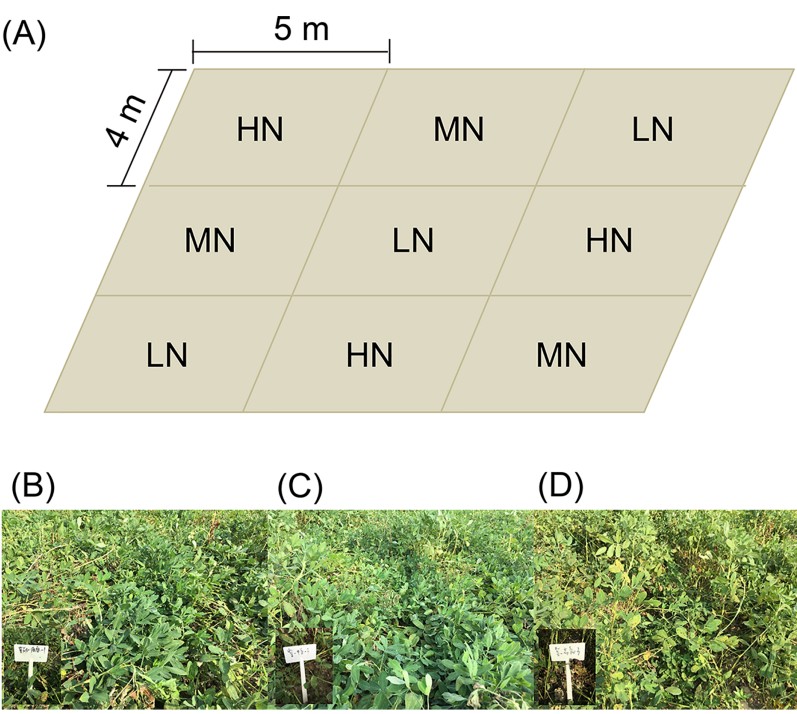

**Figure 1 Distribution diagram of each subgroup of samples (A), and photo of mature peanut from each treatment (B–D).** (B) Low nitrogen application (LH, no nitrogen); (C) middle nitrogen application (MN, 55.68 kg/hm$^2$ N); (D) high nitrogen application (HN, 111.36 kg/hm$^2$ N).

content was determined as previously described (*Chen et al., 2016*). The soil AP content was extracted with ammonium fluoride and hydrochloric acid and measured using a vis-spectrophotometer according to a previously described method (*Huang, Zhou & Liu, 2012*). The soil AK content was determined using a previously described method (*Officer, Tillman & Palmer, 2006*; *He et al., 2015*; *Li et al., 2018*). The soil exchangeable calcium (ECa) content was extracted using ammonium acetate and quantified by atomic absorption spectrophotometry (*David, 1960*; *Messmer, Elsenbeer & Wilcke, 2014*).

## Determination of rhizosphere microbiota structure

Rhizosphere soil microbiota DNA was extracted using the NucleoSpin Soil Kit (Macherey-Nagel, Düren, Germany). The concentration and quality of DNA were examined using a NanoDrop spectrophotometer (Thermo Fisher Scientific, Waltham, MA, USA) and 1.8% agarose gel electrophoresis, respectively. The hypervariable region V3–V4 of the bacterial 16S rRNA gene was amplified using bacterial universal primers 338F and 806R, as previously described (*Zhu et al., 2021*). Briefly, polymerase chain reactions (PCRs) were performed with a 10-μL reaction mix containing 1 × KOD FX Neo Buffer, 0.2 μL KOD FX Neo (1.0 U/μL), 0.4 mM of each deoxynucleoside triphosphate, 0.3 μM of each primer, and 50 ng of microbial genomic DNA. The thermal cycling procedure consisted of an initial pre-denaturation step at 95 °C for 10 min, followed by 25 cycles of 95 °C for 30 s, 50 °C for 30 s, and 72 °C for 40 s, and a final extension at 72 °C
for 7 min. The amplified products were detected by 1.8% agarose gel electrophoresis, mixed at a mass ratio of 1:1, purified using a DNA gel extraction kit (Monarch, Shanghai, China), and sequenced on a HiSeq 2500 sequencing platform (Illumina, San Diego, CA, USA) by Beijing BioMarker Technologies (Beijing, China), as previously described (*Ni et al., 2019*; *Zhu et al., 2021*).

Raw reads were merged using FLASH v1.2.11 software (*Magoč & Salzberg, 2011*) with a minimum overlap length of 10 bp, and a maximum permitted mismatch ratio of 0.2 in the overlap. The merged sequences with lengths less than 75% were filtered out using Trimmomatic version 0.33, and chimera sequences were removed using UCHIME version 8.1 (http://drive5.com/uchime/uchime_download.html). The remaining high-quality sequences were clustered into OTUs with 97% identity using USEARCH version 10.0 (https://drive5.com/usearch/). The OTUs were then phylogenetically annotated using the RDP classifier (*Wang et al., 2007*) with the SILVA version 122 database (https://www.arb-silva.de/). For alpha- and beta-diversity analyses, all samples were randomly resampled with the same number of sequences to exclude the effect of sequencing depth on the results. The alpha-diversity indices, including Chao1 and ACE indices to measure species abundance and the Shannon index to measure species diversity, were calculated using mothur v.1.30 (https://mothur.org/).

## Data analysis

Principal components analysis (PCA), redundancy analysis (RDA), and partial RDA were performed using the vegan package in R 4.2.0 (https://www.r-project.org/). The heatmap was drawn using the pheatmap package in R 4.2.0. Kruskal-Wallis rank sum test was performed and boxplots were drawn using the ggpubr package in R 4.2.0. The Galaxy platform (http://huttenhower.sph.harvard.edu/galaxy) was used to conduct the linear discriminant analysis effect size (LEfSe). Pearson correlation coefficients and bubble charts between environmental factors and rhizosphere microorganisms were calculated and drawn using the psych and reshape2 packages in R 4.2.0. Co-occurrence network analysis, based on Pearson correlation coefficients of the 82 OTUs with the highest abundances that could be identified to genus level, was performed using the igraph and psych packages in R 4.2.0. Statistical significance was set at $p < 0.05$.

# RESULTS

## Changes in soil chemical properties during peanut growth

During the experiment, although TN and TP in rhizosphere soil fluctuated, different levels of nitrogen fertilizer application did not significantly change the TN and TP at the same peanut GS (Kruskal-Wallis rank sum test, $\chi^2 = 18.32$, $p = 0.074$ for TN; $\chi^2 = 11.57$, $p = 0.397$ for TP; Figs. 2A and 2C). However, AN decreased in HN comparing with LN and MN, and this was significant at the SS and PSS (Kruskal-Wallis rank sum test, $\chi^2 = 29.51$, $p = 0.002$; Fig. 2B and Table S1). At the FPS, AP in peanut rhizosphere soil decreased first and then increased with increasing amount of nitrogen fertilizer (Kruskal-Wallis rank sum test, $\chi^2 = 21.76$, $p = 0.026$; Fig. 2D and Table S1). The change trends of TK and AK in the rhizosphere soil were almost identical, except that there were significant

differences in AK in the SS with different amounts of nitrogen fertilizer application (Kruskal-Wallis rank sum test, $\chi^2$ = 30.06, $p$ = 0.002 for TK; $\chi^2$ = 29.93, $p$ = 0.002 for AK; Figs. 2E and 2F). TK and AK first decreased and then increased with increasing nitrogen fertilizer amount (Figs. 2E and 2F). Furthermore, from the SS to the PSS, the contents of TK and AK in the soil of the three groups showed a gradual downward trend (Figs. 2E and 2F).

The pH of the rhizosphere soil also showed a similar change trend to TK and AK (Fig. 2G). ECa in rhizosphere soil showed a decreased trend in each peanut GS with increasing amount of N-fertilizer application, although this trend was not significant in most cases (Kruskal-Wallis rank sum test, $\chi^2$ = 25.721, $p$ = 0.007; Fig. 2H and Table S1).

Pearson correlation analysis showed that pH and AN (Pearson correlation coefficient = −0.621, $p$ < 0.001), AP and TP (Pearson correlation coefficient = 0.872, $p$ < 0.001), AK and TK (Pearson correlation coefficient = 0.982, $p$ < 0.001), TN and ECa (Pearson correlation coefficient = 0.482, $p$ = 0.003), AN and ECa (Pearson correlation coefficient = 0.412, $p$ = 0.012), and AK and ECa (Pearson correlation coefficient = 0.330, $p$ = 0.049) were significantly correlated. However, TN was not significantly correlated with AN (Pearson correlation coefficient = 0.266, $p$ = 0.117; Fig. S1).

## Changes of rhizosphere microbiota structure with peanut growth

A total of 2,480,563 (68,904.53 ± 1,134.57) high-quality effective sequences were obtained from 36 samples, with at least 52,488 high-quality sequences per sample. The rarefaction curve showed that the number of OTUs corresponding to the sequencing depth was close to that of the plateau (Fig. 3A), which implied that the sequencing results were representative. The alpha-diversity indices of the SSHN group were significantly lower than those of the SSLN and SSML groups at the seedling stage (Figs. 3B–3E). There was no significant difference in alpha-diversity indices at other peanut GSs (Figs. 3B–3E).

A total of 1,834 OTUs were detected, and 790 OTUs, containing 82.30 ± 0.38% of all analyzed sequences, dominated the microbiota (Table S2). PCA results showed that the samples exhibited trends of distribution with peanut GS (PERMANOVA, F = 5.475, $p$ = 0.005) and nitrogen fertilizer application (PERMANOVA, F = 1.683, $p$ = 0.025), especially peanut GS (Fig. 4A). Partial RDA showed that nitrogen fertilizer application and peanut GS explained 7.51% and 10.17% of microbial community changes, respectively (Fig. 4B), which was consistent with the PCA results (Fig. 4A). UPGMA based on the weighted UniFrac distance matrix showed that the samples were mainly clustered according to the peanut GS (Fig. 4C).

A few sequences (0.027 ± 0.004%) could not be classified at phylum level. The remaining sequences were classified into 23 phyla, of which the dominant phyla were Acidobacteria, Actinobacteria, Bacteroidetes, Chloroflexi, Cyanobacteria, Firmicutes, Gemmatimonadetes, Latescibacteria, Nitrospirae, Patescibacteria, Planctomycetes, Proteobacteria, Rokubacteria, Verrucomicrobia, and WPS-2 (Fig. 4D). The amount of nitrogen fertilizer applied had the greatest impact on the relative abundances of Bacteroidetes, Gemmatimonadetes, and Planctomycetes in the SS. In particular, high-nitrogen application significantly reduced the relative abundances of Bacteroidetes
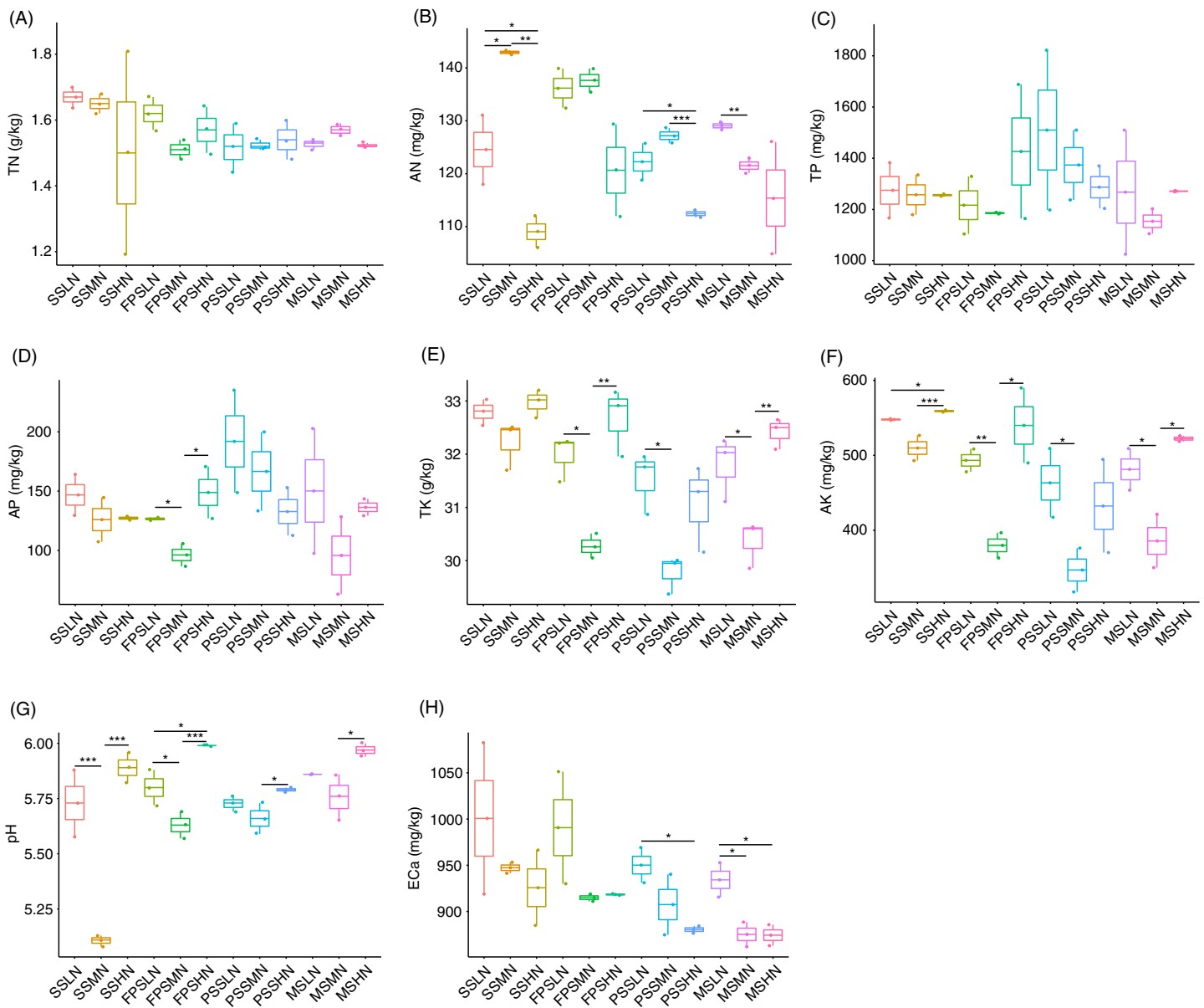

**Figure 2 Impact of nitrogen fertilizer on soil chemical properties during peanut growth.** (A) Total nitrogen (TN) content; (B) alkali-hydro-lyzable nitrogen (AN) content; (C) total phosphorus (TP) content; (D) available phosphorus (AP) content; (E) total potassium (TK) content; (F) available potassium (AK) content; (G) pH; (H) exchangeable calcium (ECa) content. SS, FPS, PSS, and MS indicate peanut seeding, flower peg stage, pod setting, and mature stages, respectively. LN, MN, and HN indicate no nitrogen, middle nitrogen (55.68 kg/hm² N), and high nitrogen (111.36 kg/hm² N) treatments, respectively. *$p < 0.05$; **$p < 0.01$; ***$p < 0.001$.

and Planctomycetes in the SS, while significantly increasing the relative abundance of Gemmatimonadetes (Figs. 4F, 4I, and 4J). Moreover, high-nitrogen application significantly increased the relative abundances of Acidobacteria and Cyanobacteria in the MS, while significantly reducing the relative abundances of Firmicutes in the MS (Figs. 4E, 4G, and 4H).

The LEfSe method was used to analyze the differences in peanut RM between peanut GSs and amounts of nitrogen fertilizer application. Our results showed that a greater

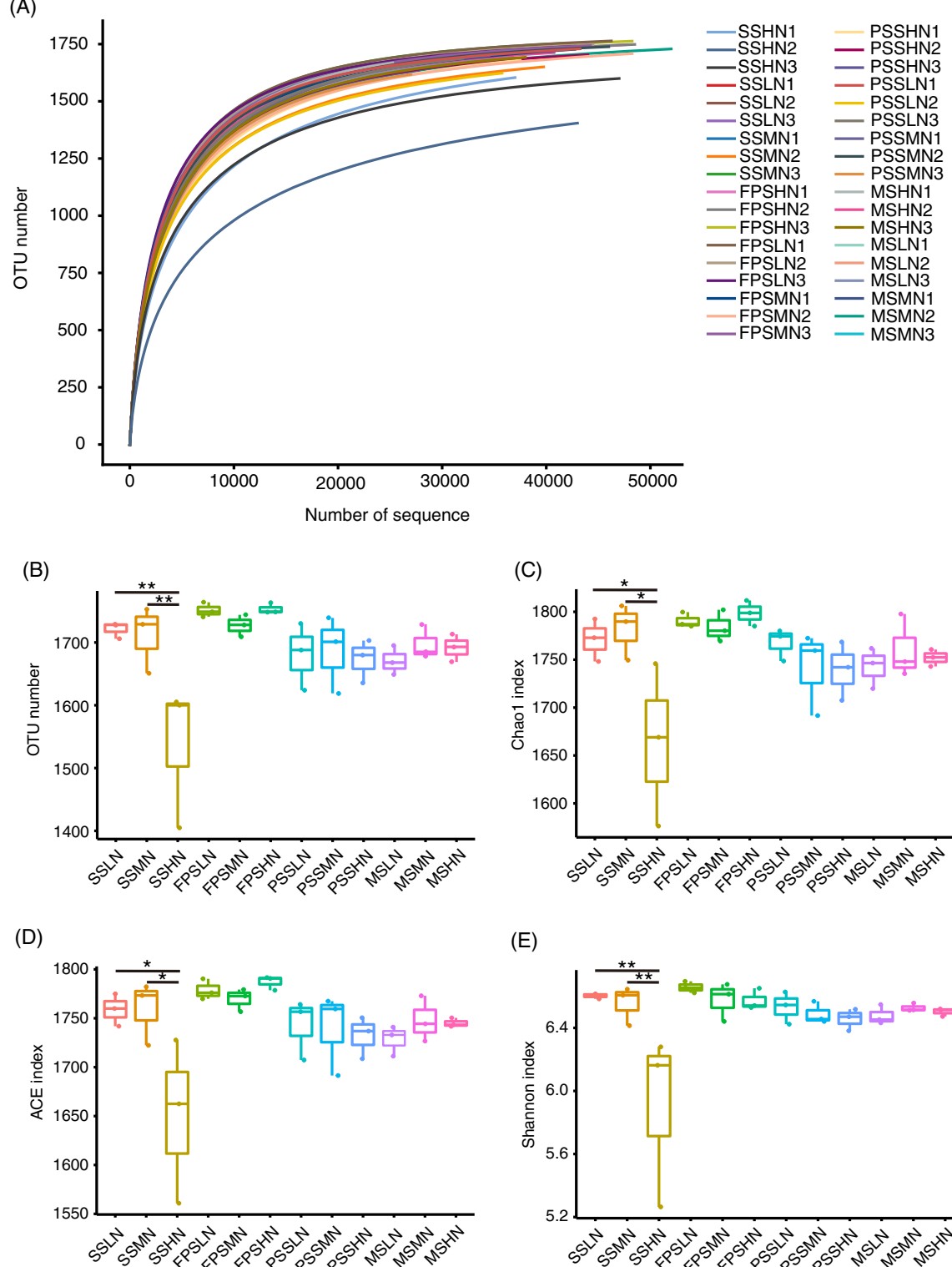

**Figure 3 Rarefaction curve (A) alpha diversity indices (B–E) of the rhizosphere microbiota of peanut.** In the sample name, SS, FPS, PSS, and MS indicate peanut seeding, flower peg stage, pod setting, and mature stages, respectively. LN, MN, and HN indicate no nitrogen, middle nitrogen (55.68 kg/hm² N), and high nitrogen (111.36 kg/hm² N) treatments, respectively. *$p < 0.05$; **$p < 0.01$.

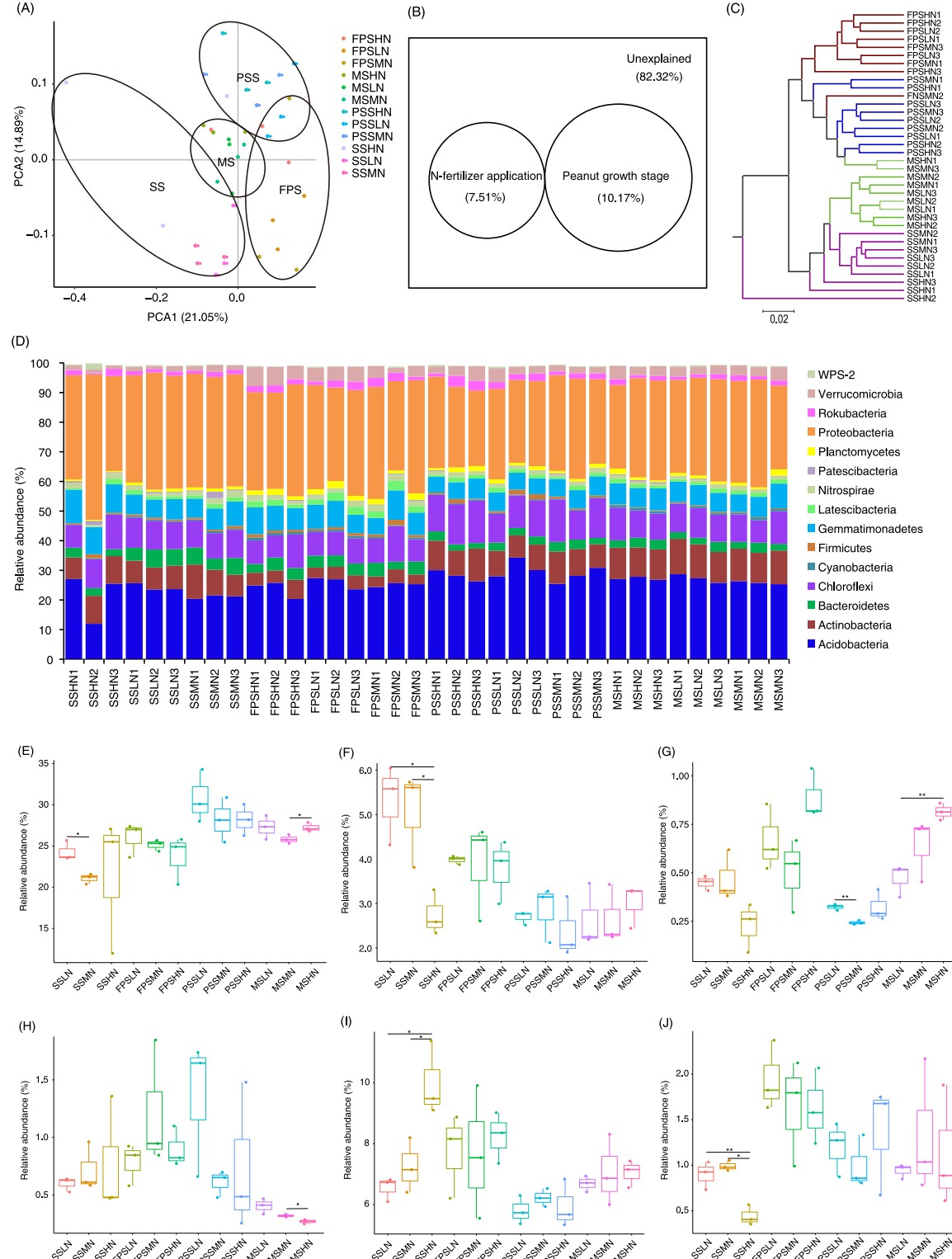

**Figure 4 Rhizosphere microbiota differences of peanut at different growth stages and with nitrogen amounts.** (A) PCA profile showing the difference of rhizosphere microbiota at OTU level, (B) Venn diagram showing the impact of N fertilizer application and peanut growth stages on rhizosphere microbiota, (C) UPGMA cluster based on weighted UniFrac distances showing the similarity of rhizosphere microbiota, (D) histogram showing the relative abundance of dominant phyla in each sample, (E) difference of the relative abundances of Acidobacteria, (F) difference of the

**Figure 4** (continued)
relative abundances of Bacteroidetes, (G) difference of relative abundances of Cyanobacteria, (H) difference of relative abundances of Firmicutes, (I) difference of relative abundances of Gemmatimonadetes, and (J) difference of relative abundances of Planctomycetes. SS, FPS, PSS, and MS indicate peanut seeding, flower peg stage, pod setting, and mature stages, respectively. LN, MN, and HN indicate no nitrogen, middle nitrogen (55.68 kg/hm$^2$ N), and high nitrogen (111.36 kg/hm$^2$ N) treatments, respectively. $^*p < 0.05$; $^{**}p < 0.01$.

number of rhizosphere bacteria species differed between peanut GSs than between the amounts of nitrogen fertilizer application (Fig. 5A; Figs. S2A and S2B). At the same peanut GS, only a small number of dominant OTUs showed significant differences in their relative abundance between the different nitrogen fertilizer application groups (Figs. S2C–S2H). In the four peanut GSs, only one OTU of *Nitrospira* showed significantly higher relative abundance in the HN group compared to the other nitrogen fertilizer groups at the SS and MS, while there was no consistent change in other OTUs. This was consistent with the results of PCA and UPGMA. The cluster analysis based on the significantly different OTUs that could be identified at the genus level also showed that samples tended to cluster according to peanut GS rather than the amount of nitrogen fertilizer application (Fig. 5B).

To determine which rhizosphere soil chemical indices significantly affect peanut RM, we used RDA to analyze the relationship between the chemical indices and RM. Since AP was significantly correlated with TP (Pearson correlation coefficient = 0.872, $p < 0.001$), and AK was significantly correlated with TK (Pearson correlation coefficient = 0.982, $p < 0.001$), and their Pearson correlation indices were greater than 0.8, we omitted TP and TK from the RDA analysis. Our results showed that AN ($R^2 = 0.515$, $p = 0.001$), pH ($R^2 = 0.255$, $p = 0.009$), AK ($R^2 = 0.324$, $p = 0.001$), and ECa ($R^2 = 0.181$, $p = 0.040$) significantly affected the peanut RM structure (Fig. 6A).

Co-occurrence network analysis showed that the relative abundances of most OTUs within a genus (such as OTUs in *Sphingomonas* and Candidatus Udaeobacter) exhibited a significant positive correlation ($p < 0.05$, Pearson correlation index > 0.6). However, there was a significant negative correlation between the relative abundances of a few OTUs from the same genus (such as OTU95 and OTU 97 in *Nitrospira*, and OTU8012 and other OTUs in Candidatus Solibacter) ($p < 0.05$, Pearson correlation index < −0.6; Fig. 6B).

Pearson correlation analysis between soil chemical indices and these bacteria showed that TP, which only significantly correlated with *Luedemannella* sp. 119-1-07, had the least correlation, followed by AP, which only significantly correlated with OTUs of *Gaiella*, Candidatus Koribacter, *Luedemannella* sp. 119-1-07, and *Bryobacter* (Fig. 6C and Table S3). The correlation patterns of AK and TK were similar; they were significantly positively correlated with OTUs of *Sphingomonas*, *Oryzihumus leptocrescens*, *Bacillus*, *Holophaga*, *Arthrobacter*, *Haliangium*, *Terrabacter*, and *Nitrospira*. However, AK and TK were significantly negatively correlated with OTUs of *Rhodanobacter*, mle1-7, PAUC26f, *Reyranella*, *Lactobacillus*, SM1A02, HSB OF53-F07, *Haliangium*, MDN1, *Bryobacter*, *Pajaroellobacter*, and *Nitrospira* (Fig. 6C and Table S3). The correlation patterns of TN and AN were dissimilar; only two OTUs of RB41 and one OTU of Candidatus Solibacter were significantly negatively correlated with both TN and AN, while most OTUs were only

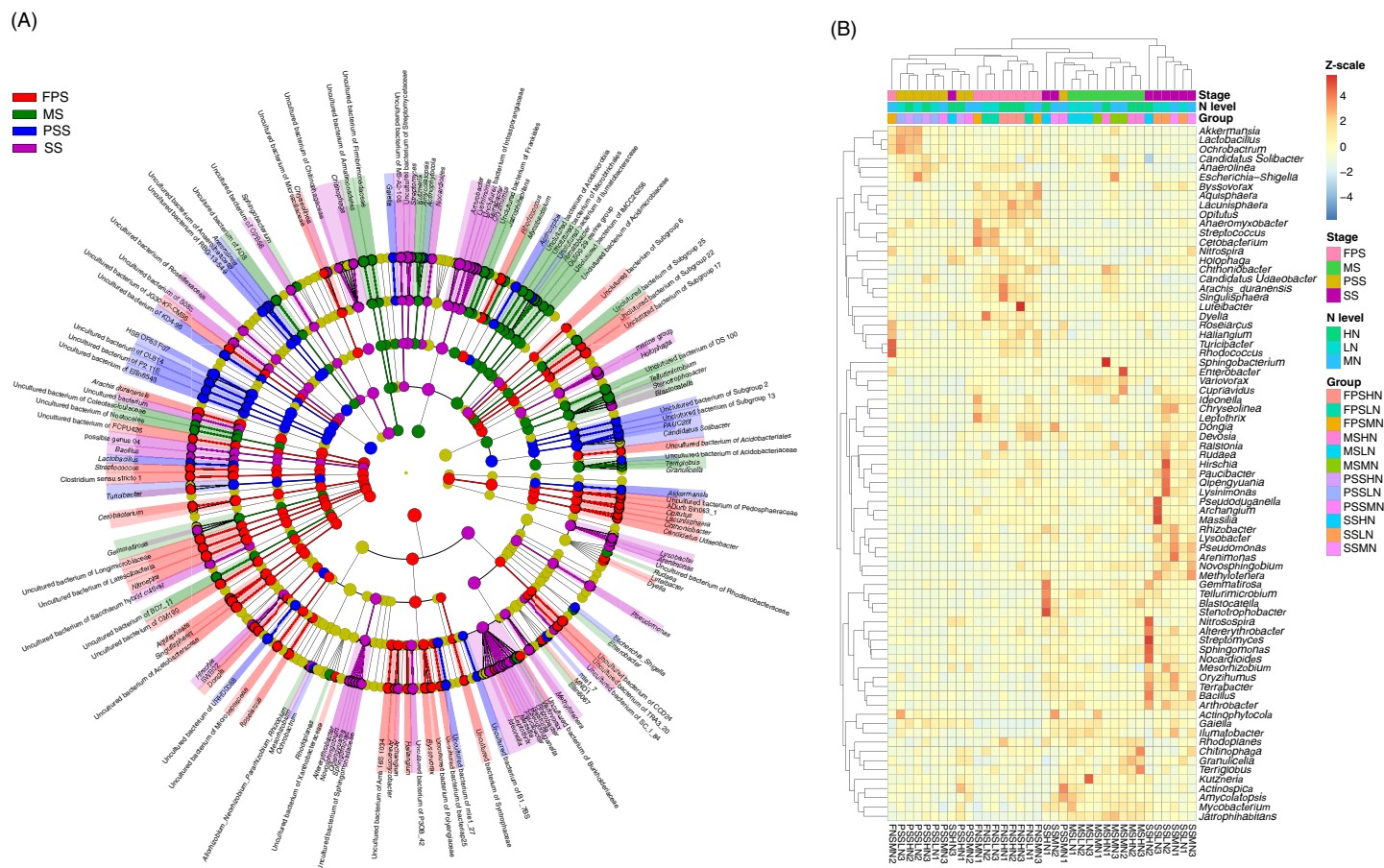

**Figure 5 Cladogram (A) and heatmap (B) profiles showing significantly different genera in rhizosphere microbiota among different peanut growth stages and amounts of N fertilizer application.** SS, FPS, PSS, and MS indicate peanut seeding, flower peg stage, pod setting, and mature stages, respectively. LN, MN, and HN indicate no nitrogen, middle nitrogen (55.68 kg/hm² N), and high nitrogen (111.36 kg/hm² N) treatments, respectively.

significantly associated with either TN or AN (Fig. 6C and Table S3). OTUs of *Gemmatimonas*, *Dongia*, *Dokdonella*, *Nitrospira*, *Phenylobacterium*, Ellin6067, *Flavobacterium*, *Bryobacter*, *Reyranella*, *Acidibacter*, *Bryobacter*, *Pseudomonas*, *Phaselicystis*, *Pseudolabrys*, *Haliangium*, *Devosia*, Candidatus Solibacter, and *Pajaroellobacter* only had a significant positive correlation with AN (*p* < 0.05; Table S3). OTUs of *Gemmatimonas*, *Bacillus*, *Rhodanobacter*, *Arthrobacter*, *Streptomyces shenzhenensis*, *Allorhizobium-Neorhizobium-Pararhizobium-Rhizobium*, *Sphingomonas*, *Mesorhizobium*, Candidatus Koribacter, and *Terrabacter* sp. Ellin109 were significantly positively correlated with TN (*p* < 0.05; Table S3). OTUs of *Nitrosospira*, Candidatus Udaeobacter, *Sphingomonas*, *Jatrophihabitans*, Candidatus Solibacter, *Nitrospira*, and *Rhodanobacter* were significantly negatively correlated with AN (*p* < 0.05; Table S3), while MND1, mle1-7, *Bryobacter*, RB41, *Gemmatimonas*, Candidatus Solibacter, and JGI_0001001_H03 were significantly negatively correlated with TN (*p* < 0.05; Fig. 6C and Table S3). One OTU of *Nitrospira* was significantly positively correlated with AN,
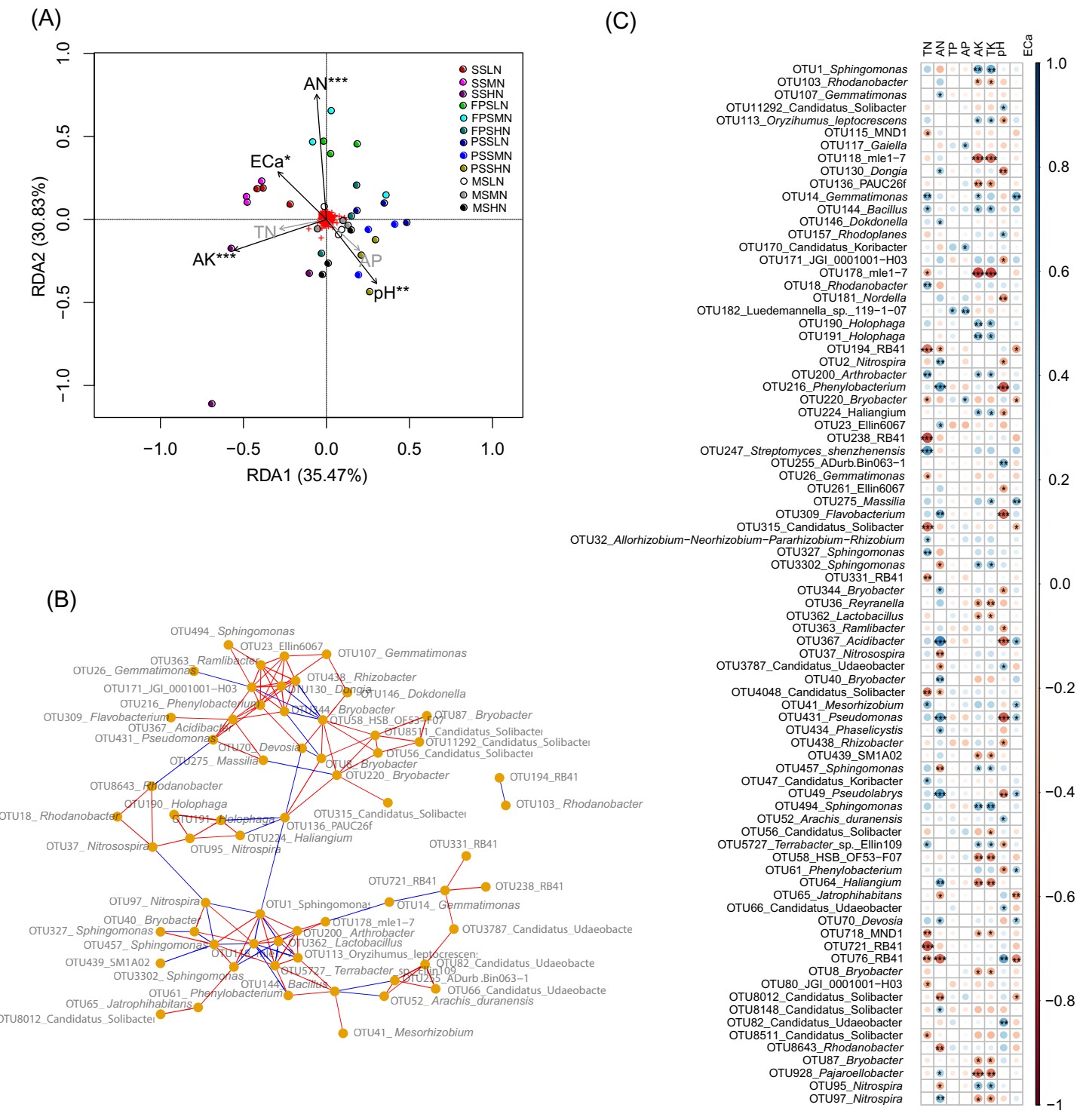

**Figure 6** RDA profile based on soil chemical indices and peanut RM (A), co-occurrence network diagram of the 82 most abundant OTUs that could be identified to genus level (B), and Pearson correlation indices of these OTUs and the soil chemical indices (C). The red and blue edges in the co-occurrence network diagram represent significantly positive correlation ($p < 0.05$ and Pearson correlation indices >0.6) and negative correlation ($p < 0.05$ and Pearson correlation indices <−0.6), respectively. SS, FPS, PSS, and MS indicate peanut seeding, flower peg stage, pod setting, and mature stages, respectively. LN, MN, and HN indicate no nitrogen, middle nitrogen (55.68 kg/hm² N), and high nitrogen (111.36 kg/hm² N) treatments, respectively. *$p < 0.05$; **$p < 0.01$; ***$p < 0.001$.

while another OTU of *Nitrospira* was significantly negatively correlated (*p* < 0.05; Fig. 6C and Table S3). This was consistent with the co-occurrence network results.

## DISCUSSION

Applying nitrogen, phosphorus, or potassium fertilizers increase the chlorophyll content, soluble protein content, and photosynthetic rate of peanut leaves, increase the activities of superoxide dismutase, peroxidase, and catlase, and finally significantly improve the pod yield of peanut. Moreover, the yield increasing effect of potassium fertilizer is greater than those of nitrogen and phosphorus fertilizers (*Zhou et al., 2007*). These results suggest that potassium may be the most important nutrient limiting factor for peanut pod yield. Our results showed that from the SS to the PSS, the contents of TK and AK in the soil of the three groups showed a gradual downward trend, while TP and AP did not exhibit this trend (Fig. 2). These results also implied that peanuts absorbed a large amount of potassium from soil during their growth. Compared with phosphorus, potassium might be the most important nutrient limiting factor for peanut pod yield.

RM plays an important role in the growth and development of plants (*Qu et al., 2020*). RM is subject to biological and abiotic factors, and its community structure can differ considerably among different plants (*Arafa et al., 2010*). Our results showed that Acidobacteria, Actinobacteria, Bacteroidetes, Chloroflexi, Cyanobacteria, Firmicutes, Gemmatimonadetes, Latescibacteria, Nitrospirae, Patescibacteria, Planctomycetes, Proteobacteria, Rokubacteria, Verrucomicrobia, and WPS-2 dominated the peanut RM. Proteobacteria are abundant in soil, among which α-Proteobacteria account for the majority (*Spain, Krumholz & Elshahed, 2019*). α-Proteobacteria contains some species that can coexist with host plants, such as *Rhizobium*, which can coexist with legumes. β-Proteobacteria play an important role in the carbon, nitrogen, and sulfur cycles in soil ecosystems. For instance, *Nitrosomonas* participates in the ammonia oxidation process in soil (*İnceoğlu et al., 2010*), and *Burkholderia* strains play an important role in symbiotic nitrogen fixation in plants (*Moulin et al., 2001*). γ-Proteobacteria contains many plant growth-promoting bacteria, which play an important role in helping plants resist pathogen infection. This group also contains *Bacillus* and *Azotobacter* bacteria with a nitrogen-fixing function (*Ge & Sun, 2020*). In peanut RM, we found that Acidobacteria had the same abundance as Proteobacteria (Fig. 4D). Functions of Acidobacteria include degrading plant residues, participating in single carbon compound metabolism, and participating in the iron cycle (*Wang et al., 2016*). Actinobacteria species are mostly saprophytic heterotrophic bacteria, which can absorb and utilize a variety of organic compounds and play an important role in the material cycle in nature (*Passari et al., 2017*). Actinobacteria species can also produce a large number of antibiotics and secondary metabolites for clinical and agricultural applications (*Zhao et al., 2018*). In addition, Brunchorst exists in the root nodules of non-leguminous woody plants for symbiotic nitrogen fixation (*Laws & Graves, 2005*; *Sun et al., 2020*). Our results showed that application of 111.36 kg/hm² nitrogen fertilizer significantly increased the relative abundance of Gemmatimonadetes, and decreased the relative abundances of Bacteroidetes and Planctomycetes. Application of 55.68 kg/hm² nitrogen fertilizer significantly reduced the relative abundance of

Acidobacteria at the SS (Fig. 4). In other peanut GSs, only the relative abundance of Cyanobacteria was significantly reduced in the MN group at the PSS and increased in HN at the MS compared with the LN group (Fig. 4). These results implied that nitrogen fertilizer application only affected the composition of peanut RM at the SS, and that its effect on RM decreased after the SS, which may be due to the gradual self-healing of the soil microenvironment in response to short-term nitrogen input and tends to be homogenized over time. The differences in RM structure were mainly reflected in the different peanut GSs (Figs. 4 and 5).

The species and quantity of root exudates of crops in different GSs are different, so it follows that the RM, which uses root exudates as energy material, will also change in different GSs. This has been observed for the RM of other plant species (*Tkacz et al., 2015*). *Tong, Gao & Jin (2019)* reported that the richness and diversity of RM was highest in the budding stage of strawberry, and that the relative abundance of Proteobacteria increased, while Acidobacteria decreased gradually with strawberry growth. *Yan et al. (2020)* reported that the RM structure changed significantly in the vegetative growth stage of potato, then gradually stabilized at the beginning of the potato setting stage, and changed considerably again at the later tuber maturity stage.

Various interactions exist between RM and soil nutrients (*Arafa et al., 2010*). Nitrogen-fixing bacteria in soil can fix $N_2$ for plants (*Kuypers, Marchant & Kartal, 2018*), and phosphate and potassium dissolving bacteria can secrete organic acids or extracellular enzymes to promote the dissolution of insoluble phosphate and aluminosilicate and input effective elements into the soil (*Umamaheswari & Murali, 2010*; *Liu et al., 2020*). However, the RM structure is also affected by the soil nutrients (*Arafa et al., 2010*; *Igiehon & Babalola, 2018*). The application of exogenous fertilizer can cause changes in the rhizosphere microenvironment, in which bacteria are sensitive to changes in soil nutrients (*Igiehon & Babalola, 2018*). *Sang, Zhao & Zhang (2020)* found that application of organic liquid fertilizer increases the available nutrients and organic matter of soil, and AN, organic matter, and AP have an important impact on RM. *Chen et al. (2021)* found that biomass phosphorus was an important factor affecting the change in RM structure in the Hubei experimental area, and ammonium nitrogen significantly affected RM structure in the Zhejiang experimental area. Our results showed that pH, AN, AK, and ECa significantly impacted the peanut RM structure in the southern paddy soil environment. It has been widely reported that pH significantly affects RM structure (*Lauber et al., 2009*; *Cho, Kim & Lee, 2016*; *Yao et al., 2017*; *Liu, Li & Yao, 2021*). Urea is converted into $NH_4^+$ under the specific catalysis of soil urease. Nitrobacteria can convert $NH_4^+$ into $NO_2^-$, release a large amount of $H^+$, reduce soil pH, and then affect the soil microenvironment. Although AK significantly affected RM structure, it was also significantly related to TK. Therefore, further experiments are required to determine whether the correlation between AK and RM was caused by the impact of TK on RM.

Our previous experiments showed that nitrogen fertilizer application significantly increased peanut yield (*Yang et al., 2021*). However, the results of the present study showed that nitrogen fertilizer application had little effect on peanut RM, and the effect was obvious only at the SS. This was despite the significant influence of nitrogen fertilizer

application on the soil TK and AK contents, and the significant correlation between AK and the peanut RM composition. In the four peanut GSs, only one OTU of *Nitrospira* showed significantly higher relative abundance in the HN group compared to the other nitrogen fertilizer groups at the SS and MS. These results imply that the changes in peanut RM at different GSs may be affected by soil factors and/or peanut secretions that we did not detect. Moreover, *Nitrospira* might be used as a potential beneficial bacterium to promote peanut growth.

*Pseudomonas stutzeri* A1501 is the most studied N-fixing endophyte. It enhances the remediation capacity of broad bean plants and also helps to control plant pathogens (*Tkacz & Poole, 2015*; *Igiehon & Babalola, 2018*). *Rhizobium* species significantly increase the height, pod number, length, and seed weight of *Vigna mungo* and *Vigna radiate* (*Igiehon & Babalola, 2018*). We found that the relative abundances of *Pseudomonas* and *Rhizobium* showed significant positive correlations with AN and TN, respectively. Although *Azoarcus* species and *Herbaspirillum seropedicae* enhance host plant growth by fixing N (*Igiehon & Babalola, 2018*), our results showed that they did not dominate the peanut RM. This implies that rhizosphere bacteria identified as promoting host plant growth using the traditional culture method may not have the same effect in the peanut RM in the field. This may be due to the high TN content in our experimental field, which may have limited the advantage conferred by the N-fixing bacteria. Moreover, soil high N background probably weakened the effects of N application on rhizosphere bacterial diversity. However, the effects should be further experimentally verified.

## CONCLUSIONS

The results of the present study showed that nitrogen fertilizer application had little effect on peanut RM, and this effect was evident only at the SS. In the four peanut GSs, only one OTU of *Nitrospira* significantly increased in relative abundance with HN at the SS and MS, while there was no consistent change in other OTUs. The difference in RM between different peanut GSs was greater than that caused by the amount of nitrogen fertilizer. This may be due to the substantial changes in soil chemical properties at different peanut GSs, especially in AN, pH, and AK (or TK), as these significantly affected the RM structure.

## ACKNOWLEDGEMENTS

We would like to thank Dr. Jiajia Ni at Guangdong Meilikang Bio-Science Ltd., China for assistance with data analysis.

### Funding

This research was funded by the National Key R & D Program of China, grant number 2018YFD1000902. The funders had no role in study design, data collection and analysis, decision to publish, or preparation of the manuscript.

## Grant Disclosures

The following grant information was disclosed by the authors:

National Key R & D Program of China: 2018YFD1000902.

## Competing Interests

The authors declare that they have no competing interests.

## Author Contributions

- Zheng Yang conceived and designed the experiments, performed the experiments, analyzed the data, prepared figures and/or tables, and approved the final draft.
- Lin Li performed the experiments, analyzed the data, prepared figures and/or tables, and approved the final draft.
- Wenjuan Zhu performed the experiments, prepared figures and/or tables, and approved the final draft.
- Siyuan Xiao performed the experiments, authored or reviewed drafts of the article, and approved the final draft.
- Siyu Chen performed the experiments, authored or reviewed drafts of the article, and approved the final draft.
- Jing Liu performed the experiments, prepared figures and/or tables, and approved the final draft.
- Qian Xu performed the experiments, prepared figures and/or tables, and approved the final draft.
- Feng Guo conceived and designed the experiments, analyzed the data, authored or reviewed drafts of the article, and approved the final draft.
- Shile Lan conceived and designed the experiments, analyzed the data, authored or reviewed drafts of the article, and approved the final draft.

## DNA Deposition

The following information was supplied regarding the deposition of DNA sequences:

All sequences are available at NCBI SRA: PRJNA773647.

## Data Availability

All sequences are available at NCBI SRA: PRJNA773647. The raw data are available in the Supplemental Files.

## Supplemental Information

Supplemental information for this article can be found online at http://dx.doi.org/10.7717/peerj.13962#supplemental-information.

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
