# Peer review of "Nitrogen fertilizer amount has minimal effect on rhizosphere bacterial diversity during different growth stages of peanut"

_PeerJ, doi:10.7717/peerj.13962_

## Round 0.1 · original submission · Major Revisions

Line 82, what does 'duration' mean here, needs to be clarified?
In Suppl table 1, please provide the units of soil traits measured.
While preparing the manuscript, the authors must type it in double spacing


Reviewer 1 ·

Basic reporting

Authors have tried to study the impact of nitrogen fertilizer application on the structure and diversity of rhizosphere microbiota (RM) at different growth stages (GSs) of Peanut plant. Authors have applied different concentration of Nitrogen fertilizers at the different growth stages and studied the microbial diversity through 16s rRNA sequencing. Authors have also chemically profiled the soil used in the study.

There are following suggestions which could be considered :
The language of the article needs to be improved.
Isoflavones in introduction has been described without any context to previous explanation. Authors should have included some points to connect Isoflavones with Peanut growth.
Line 82 (RM and their duration have not been fully described). Incorporating some references will make the statement more impactful.
Some of the literature references are very old, authors are required to incorporate some recent references, along with the old ones.

Experimental design

The experimental design (Line 95 to 106) needs to be rewritten with proper experimental objectives.
Data Analysis should be written more elaborately with the proper input values used for the analysis. Versions of software programs and the links for used online platforms should be provided in Data analysis.

Validity of the findings

Authors have described the changes in soil in detail. Although there are some suggestions to add, For example: Under the heading “Changes in soil chemical properties during peanut growth” the AN is showing significant decrease with the HN condition in all the groups. Authors should rewrite that result. The results have not been interpreted according to the graphs shown in Fig 2. TK and AK shown an increment at HN levels which has not been considered by the authors.
Fig 2 graphs needs to be presented with the alternative background or plain background with colored box plots. Authors could also increase the line width of box plots. The arrangement of the groups could be represented as from LN to HN or vice versa (HN, MN and LN). The groups are not following an increasing or decreasing order.
In Fig 3b and 3C, X axis of the graphs representing groups is needed to be rearranged in order. (LN, MN, HN) or vice versa.
In the results section while describing Fig 4A, the PCA needs to be inferred and explained in more detail.
Discussion could be improved. Currently, it reads like the compilation of results.
Authors could also discuss about:
The molecular pathways that would have got impacted with the nitrogen supplementation or deficit. What led to the increment or the decrement of the relative abundances of microbial population, and how did it impact the different GS of peanut plants.
Authors could also state about how their results are in comparison to other findings and could also discuss some state-of-the-art research.
At the end of discussion, authors could discuss about the implication of their finding to promote peanut production. How their results could be benefited to enhance the peanut quality or production.

Additional comments

Line 35 OTU please elaborate the acronym.
Fig. S2 could be merged with Fig 4 D, to get the overall change at single glance.
Fig S3, very low resolution could not read the text. Author should provide high resolution images.
Fig 5 and 6 are of low resolution. Both the figured could be merged.

Reviewer 2 ·

Basic reporting

This manuscript describes an investigation into the effect of nitrogen on rhizosphere bactrical diversity during different growth stages of peanut. The research topic is within the scope of PeerJ. However I suggest the author further indentify what knowledge gap is filled by this research, such as the interaction among N application, legume and nitrogen-fixing bacteria.

Experimental design

As we know N, especially nitrate-N, is prone to movement in soil. The annual precipitation was so high as 1361.6 mm and plot area was only 4 m5 m. The methodology how to avoid N movement among plots of different N treatment should be described clearly. In addition, the soil N content is rather high. It is wondered that if soil high N background diminishes the effects of N application to rhizosphere bacterial diversity?

Validity of the findings

I do not think it is make any sense to analysis Person correlation between items of soil chemical properties (Fig S1)

Additional comments

1. The manuscript involves rhizosphere bacterial diversity only. The title of manuscript should match the content.
2. Line 42, please describe the volume of each chemical reagent
3. Line 89, How did “annual accumulated temperature of 5457 °C” was calculated? more than 0oC, or 10 oC?
4. Line 156. Please explain the version of SILVA
5. Please standardize N treatment order in explanatory note and in Figs. For example, the N treatments are LN, MN and HN in explanatory of Fig 2 & 3; whereas the order is SSHN, SSLN and SSMN

---

## Round 0.2 · Minor Revisions

The authors have improved the manuscript however, a minor revision is still needed to improve the readability of the manuscript.

Reviewer 1 ·

Basic reporting

Authors have incorporated the suggested changes and significantly revised the manuscript. In the Results section, authors need to rewrite the section "Changes in soil chemical properties during peanut growth"
especially from line 186-194. The language needs to be improved in these lines, currently the results described in this paragraph are confusing. It should be rewritten.

There is still the cope of language improvement in the manuscript, more professional/scientific language could be used, especially in materials and results section.

Experimental design

No Comment

Validity of the findings

No Comment

Additional comments

No Comment

Reviewer 2 ·

Basic reporting

Author(s) have made great efforts to modify the paper. Although the title of paper is clear the scientific mechanism would be clarified further.

Experimental design

Line 39: Abbreviation of AN, AK and TK was avoided and full name are given at the first time in the paper.
Line 82: Is “growth stage” better than “duration”?
Line 95: Please express the units of alkali-hydrolyzable nitrogen, available phosphorus, and available potassium as mg/kg.
Line 98-99 & Figure 2. I suggest that N levels are expressed as N rate, rather than urea application rate. If so it would be consistent with P and K as well.

Validity of the findings

Line 192-194: Please explain the influences of N rate on TK and TP in suitable section.
Line 379-380: The relationships between Bacillusa bundance between AK and TK and the conclusion following is not related to the topic of the paper

Additional comments

No comments

---

## Round 0.3 · accepted · Accept

The authors have made significant corrections as suggested by all the reviewers.